# Late Eating Is Associated with Obesity, Inflammatory Markers and Circadian-Related Disturbances in School-Aged Children

**DOI:** 10.3390/nu12092881

**Published:** 2020-09-21

**Authors:** Nuria Martínez-Lozano, Asta Tvarijonaviciute, Rafael Ríos, Isabel Barón, Frank A. J. L. Scheer, Marta Garaulet

**Affiliations:** 1Department of Physiology, University of Murcia, IMIB-Arrixaca, 30120 Murcia, Spain; nuria_coy@hotmail.es; 2Interdisciplinary Laboratory of Clinical Analysis (Interlab-UMU), University of Murcia, 30120 Murcia, Spain; asta@um.es; 3Health Area of Lorca, Lorca, 30800 Murcia, Spain; rrdemoya@yahoo.es (R.R.); Ibmizzi@hotmail.com (I.B.); 4Division of Sleep Medicine, Harvard Medical School, Boston, MA 02115, USA; 5Medical Chronobiology Program, Division of Sleep and Circadian Disorders, Brigham and Women’s Hospital, Boston, MA 02115, USA

**Keywords:** obesity, children, dinner, chronobiology, biomarkers

## Abstract

Late eating has been shown to promote metabolic dysregulation and to be associated with obesity in adults. However, few studies have explored this association in children. We compared the presence of obesity, metabolic alterations and circadian-related disturbances between school-aged children who were early dinner eaters (EDE) or late dinner eaters (LDE). School-age children (*n* = 397; 8–12 years; mean BMI (range): 19.4 kg/m^2^ (11.6–35.1); 30.5% overweight/obesity) from Spain were classified into EDE and LDE, according to dinner timing (Median: 21:07). Seven-day-dietary-records were used to assess food-timing and composition. Non-invasive tools were used to collect metabolic biomarkers (saliva), sleep and circadian-related variables (body-temperature and actigraphy). Compared to EDE, LDE were more likely to be overweight/obese [OR: 2.1 (CI: 1.33, 3.31); *p* = 0.002], and had higher waist-circumference and inflammatory markers, such as IL-6 (1.6-fold) (*p* = 0.036)) and CRP (1.4-fold) than EDE (*p* = 0.009). LDE had alterations in the daily patterns of: (a) body-temperature, with a phase delay of 26 min (*p* = 0.002), and a reduced amplitude (LDE = 0.028 (0.001) and EDE = 0.030 (0.001) (Mean (SEM); *p* = 0.039); (b) cortisol, with a reduced amplitude (LDE = 0.94 (0.02) and EDE = 1.00 (0.02); *p* = 0.035). This study represents a significant step towards the understanding of novel aspects in the timing of food intake in children.

## 1. Introduction

According to the World Health Organization, the prevalence of obesity among children has risen dramatically since 1975, reaching up to 18% of the population [1]. Nevertheless, successful treatments of obesity in children remain a challenge [2] and underlying causes should be investigated more deeply, in order to improve the prevention of obesity at these ages.

Chrono-medicine is receiving increasing attention because of the already demonstrated association between the timing of behaviours, such as the timing of food intake, physical activity or sleep, circadian alterations and pathology [3,4,5,6]. However, circadian health in children is less studied than in adults, due to the difficulties associated with the need for repetitive samplings to assess 24-h variations in metabolic parameters. In order to evaluate circadian health without causing stress, non-invasive methods are being proposed. Some examples are wrist devices that measure daily rhythms in temperature or physical activity [7] or saliva samples, instead of blood, to assess daily rhythms of circadian hormones such as cortisol. These approaches are easy-to-obtain, safe, stress-free and economic [8].

Delayed timings of food intake and sleep have been related to obesity, metabolic dysregulation and increased values of inflammatory markers [9,10,11]. In adults, late eating, which refers to a delay in the timing of meals (usually the main meal of the day or the last meal, as dinner) is robustly associated with hyper-glycemia [12], impaired glucose tolerance [13] and increased risk of poor cardio-metabolic health [14]. Late eating has also been associated with worse sleep patterns [15] and late bedtimes in adults [16].

In children, night eating has been related to increased Body Mass Index (BMI) [17], but results are not consistent across studies, and no association between late dinner and obesity has been detected in a population of UK children [18]. Furthermore, no studies have been performed on late eating and obesity, including inflammatory markers and circadian-related parameters.

In the current study, we aimed to study whether late dinner eaters were more prone to suffer from obesity, metabolic disturbances and circadian disruption than early eaters in a school-age population. Our hypothesis is that late dinner eaters (LDE) will show higher obesity, increased inflammatory markers and circadian-related disturbances than early dinner eaters (EDE).

## 2. Materials and Methods

### 2.1. Subjects

School-aged children (8–12 years) from the Obesity, Nutrigenetics, Timing, and Mediterranean Junior study (ONTIME-Jr; ClinicalTrials.gov ID: NCT02895282) were recruited from three schools in a Mediterranean area of Spain during the years 2014 to 2016. To select the children, school board was contacted, and a briefing was shown about the project in order to inform the steering committee of the school. Out of a total of 432 children, only 397 were included in the study, because their questionnaires were fully completed with regards to food quantities and timing (7-day dietary record). After obtaining approval, parents were convened for an informative rendezvous about the study and a written consent to participate was provided. At the end of the study, a full written report on the circadian health of each child was handed over. The study was authorized by the Ethics Committee of the University of Murcia (ID: 1868/2018). All procedures performed were in accordance with the ethical standards of the institutional and national research committee and with the 1964 Helsinki declaration and its later amendments or comparable ethical standards. Recruitment procedure and methodology have been previously described [19].

### 2.2. Classification of Late (LDE) and Early (EDE) Dinner Eaters

In order to classify the children in late and early dinner a 7-day dietary record (a paper-and-pencil version) including food quantities and timing was completed by the children [20]. Children were classified into two groups according to the median of the dinner time 21:07. Children who had dinner before 21:07 were considered early dinner eaters (EDE), while those who had dinner after 21:07 were late dinner eaters (LDE). Midpoint of food intake was defined as an average of the seven days of the midpoint between breakfast and dinner times (first and last eating episode). In addition, we calculated the following variables: (A) Social jet lag of dinner timing: the difference between dinner timing on weekends and on weekdays. (B) Social jet lag of midpoint of food intake: the difference between the midpoint of food intake on weekends and weekdays. We also defined intraindividual variation variables, such as: (C) Dinner phase deviation: the standard deviation of the mean of dinner timing. (D) Midpoint of food intake phase deviation: the standard deviation of the mean of midpoint of food intake. (E) Interday phase change (in dinner timing and midpoint of intake): $\sqrt{{\left(ai-bi\right)}^{2}}$ (*ai* = dinner timing or midpoint of intake on day *i*; *bi* = dinner timing or midpoint of intake on previous day).

The 7-day dietary records, including food quantities and timing, were completed daily by the children with their parents’ help. They were also informed that they had to weigh the food and write it down every day. Furthermore, the staff was in constant communication with the parents through a cell phone app for any enquiries or concerns that could arise during the intervention week. The diets were coded by a trained dietitian and reviewed by a senior nutritionist, and total energy intake and macronutrients composition were analysed with a nutritional evaluation software program (Grunumur 2.0 8) [21] based on Spanish food composition tables [22].

### 2.3. Anthropometric Measures and Body Composition

Measurements such as BMI and waist circumference were performed on the first day of the week of the study and at the same time of day in the morning, as previously described [19]. Body weight was assessed in barefoot subjects wearing light clothes using a digital scale accurate to the nearest 0.1 kg. Height was determined using a portable stadiometer (rank, 0.14–2.10). Children were positioned upright, relaxed and with their head in the Frankfort plane. BMI was calculated according to the formula: weight (kg)/height^2^ (m^2^). Children were categorised into normal weight and overweight/obese according to the sex-and-age specific BMI cut-points proposed by the International Obesity Task Force [23]. BMI and age- and sex-specific z scores were calculated using WHO Growth [24]. Waist circumference was measured in standing position with their feet together at the midpoint between the last rib and the crest of the ilium (top of the hip bone). Measurements were done after participants placed their arms at their side with the palms of their hands facing inwards and breathing out gently. Data were recorded to the nearest 0.1 cm using a metallic tape. Total body fat was determined by bioelectrical impedance, using TANITA TBF-300 (Tanita Corporation of America, Arlington Heights, IL, USA) equipment.

### 2.4. Sleep

Children (together with their parents) completed 7-day sleep diaries adapted for the appropriate age group [25] which included: (a) time in bed (bedtime, number of awakenings during sleep and get up time); (b) time and duration of naps [20]. Furthermore, sleep duration was objectively determined by a formula integrating the objectively measured motor activity, body position and skin temperature, as previously described [26].

### 2.5. Chronotype

An age-appropriate Spanish version of the Munich Chronotype Questionnaire (MCTQ) was used [27].

### 2.6. Saliva Determinations

In order to measure several metabolic and inflammatory markers, saliva samples were collected in a standard centrifugation tube with a small cotton swab inside (Salivette; Sarstedt, Barcelona, Spain). Samples were obtained the same day of the week (Sunday) for all the children, at home and with their parents help, as previously used in adults [28]. Moreover, the saliva collection was performed before breakfast at 09:00 (*n* = 129) in fasting conditions. All samples were maintained refrigerated at 4 °C until delivered (one day) to the laboratory and then stored at −80 °C until analysed. For cortisol, three salivary samples were collected: one before breakfast (9:00), one before lunch (14:00) and one after dinner (23:00), and were measured by radioimmunoassay (IZASA, Barcelona, Spain). Triglycerides were quantified by a method based on a series of coupled enzymatic reactions (Beckman Coulter Ireland Inc., Co., Clare, Ireland). Glucose was determined by employing a hexokinase-based methodology (Beckman Coulter Ireland Inc., Co., Clare, Ireland). The two parameters were analysed in automated biochemistry analyser (Olympus AU600, Olympus Diagnostical GmbH, Freiburg, Germany). Interleukin (IL) 1β, IL-6, IL-8, insulin, leptin, monocyte chemotractant protein 1 (MCP-1), nerve growth factor (NGF), hepatocyte growth factor (HGF), tumour necrosis factor alpha (TNFα) were analysed using commercially available kits (MILLIPLEX MAP Human Adipokine Magnetic Bead Panel 2-Endocrine Multiplex Assay; Life Science, Darmstadt, Germany), according to the manufacturer’s instructions. C-reactive protein (CRP) was analysed using commercially available kits (MILLIPLEX MAP Human-CRP Assay; Life Science, Darmstadt, Germany) according to the manufacturer’s instructions.

### 2.7. Activity and Temperature Variables

Children wore a wristwatch for 7 days of the study on their non-dominant hand, that integrated two different sensors: (A) a tri-axial accelerometer sensor that measures average physical activity and programmed to record data every 30 s; (B) a temperature sensor for determining wrist temperature rhythms previously validated [29,30] that quantifies information every 5 min [26]. Motor activity was expressed as the accumulative changes in three-axis tilt with respect to the previous point and expressed as degrees per minute. To assess sleep objectively, body position was calculated as the angle between *X*-axis of the accelerometer and the horizontal plane, where 0° represents the arm in a horizontal position and 90° when vertical. Motor activity determinations have been previously validated with the commonly used wrist-worn Actiwatch accelerometer, and almost all the parameters showed high correlations between both devices [31]. Because the device used in the current study integrates the activity sensor together with a temperature sensor, we were able to use both the motor activity and temperature readings to estimate wear time. Non-wear time was defined as an interval in which motor activity readings were lower than 4°/min and the skin temperature readings were out of the physiological range (i.e., <28 °C). A valid day was defined as a day with at least 16 h of wear time.

To characterize circadian rhythms (24 h), Cosinor’s analysis was used. Circadian parameters such as relative amplitude (i.e., difference between the maximum (or minimum) value of the cosine function and mesor); and acrophase (i.e., time at which the peak of a rhythm occurs) were calculated. These rhythmic parameters were obtained using an integrated package for temporal series analysis Kroniwizard (https://kronowizard.um.es/kronowizard) (Chronobiology Laboratory, University of Murcia, Murcia, Spain, 2015).

### 2.8. Statistical Analysis

In the population studied, the homogeneity of variances was verified with the Levene test. In those normal distributed variables, such as age, BMI, waist, total energy intake, midpoint of food intake, dinner duration (h), time in bed (h), chronotype (MCTQ) and average physical activity level, differences between LDE and EDE were analysed by ANCOVA, and logistic regression analyses were used to test differences in overweight and obesity between EDE and LDE. Differences between sexes were further explored. Analyses were adjusted for schools, age, academic year, sleep (time in bed and sleep duration) and sexes, when necessary. Odd ratio for the association of obesity with timing of food intake was adjusted for schools, age, academic year and sex. To assess differences in frequencies in overweight/obesity, and chronotype between LDE and EDE, chi-square tests were used. For non-normal distributed variables, such as the inflammatory biomarkers, non-parametric tests, such as Mann–Whitney U test, were used. Salivary morning-evening cortisol ratio values were log-transformed. All statistical analyses were performed using SPSS version 20.0 (SPSS, Chicago, IL, USA). A two-tailed *p*-value of <0.05 was considered as statistically significant, and a *p*-value between 0.05 and 0.10 was considered as a trend..

## 3. Results

A total of 397 children with ages between 8 and 12 years were included in the study. General characteristics of participants, timing of food intake and energy intake distribution across meals are summarized in Table 1. Children had dinner at 21:07 (CI: 20:08; 22:06) (odds ratio (95% confidence interval) and dinner energy intake constituted the 28% of the total energy of the day.

No statistically significant differences were found in the distribution of energy intake across meals between LDE and EDE. Only for breakfast, the percentage of daily energy was significantly lower in LDE than in EDE *(p* = 0.002) (Figure 1A). Furthermore, LDE had lower levels of cortisol in the morning (*p* = 0.044) (Figure 1B) and a more reduced amplitude (morning–evening cortisol difference: LDE = 0.94 (0.02) and EDE = 1.00 (0.02); *p* = 0.035) than EDE.

Late dinner eaters (LDE) had higher BMI and waist circumference than EDE (Table 1) and accounted for a higher proportion of overweight/obese children than early eaters (*p* < 0.05) [OR = 2.1, (CI: 1.33, 3.31)]. Association between LDE and BMI remained significant when adjusted for objective sleep duration (*p* = 0.019) and the same trend was obtained when adjusted for time in bed (*p* = 0.055).

When standardizing by sex, these associations remained significant only in girls (Appendix A). No significant associations were found between intraindividual variation variables (of dinner timing and midpoint of food intake) and BMI z-scores (Appendix A).

Social jet lag of dinner timing and midpoint of food intake differed across school grades, with grade 2 (10–11 years) having less social jet lag than grades 1 (8–9 years) and 3 (11–12 years). No significant differences were found in social jet lag values between both sexes (Appendix A).

Late dinner eaters (LDE) had higher values of CRP (1.4-fold) and IL6 (1.6-fold) than EDE (Table 2). However, no significant differences were found for glucose, insulin, triglycerides, leptin, IL1, IL8, TNFα, etc.

Dinner energy content and macronutrient distribution were similar between LDE and EDE. However, the duration of dinner was shorter among late eaters (Table 3). In addition, LDE presented a later chronotype with later sleep centre and a shorter time in bed than EDE.

Results from wrist peripheral temperature showed a phase delay of 26 min, among the LDE as compared to EDE (Acrophase for LDE = 03:42 (00:05); and EDE = 03:16 (00:05); (mean (SEM), *p* = 0.002) (Figure 2A), and significantly lower values in the relative amplitude of the rhythm in LDE than in EDE (LDE = 0.028 (0.001); (AU) and EDE = 0.030 (0.001) (AU), *p* = 0.039) (Figure 2B). Moreover, wrist temperature values differed between LDE and EDE in the morning (from 09:30 to 11:00 and from 11:30 to 12:00) and in the evening (16:00 to 17:30) (Figure 2A). After dinner, in the postprandial hours (3 h), LDE had a delay of 20 min in the increase of temperature (*p* = 0.005) and lower temperature values (LDE = 33.29 (0.07) °C; and EDE = 33.57 (0.07) °C; *p* = 0.005) than EDE.

## 4. Discussion

The current work suggests that the timing of dinner is a relevant factor in obesity for school-age children. Late dinner eaters were 2.1-times more likely to be overweight/obese than early eaters, and had significantly higher values of well-known inflammation markers such as IL6 and CRP. This study provides a circadian-related view of several physiological alterations that associate to a late dinner, since late eaters suffered modifications in the daily pattern of body temperature with a 26 min phase delay and reduced amplitude; and in the daily pattern of cortisol, with reduced amplitude (i.e., lower morning-evening cortisol ratio).

Studies on meal timing in children are scarce, and some divergences exist in paediatric obesity-related literature regarding the timing of the main meals of the day and the potential impact on obesity and metabolic alterations. In agreement with our current results that show higher BMI z-score and waist circumference in late eaters than in early eaters, some authors have observed that late eating associates with obesity in different age groups of children [17]. However, other authors do not find this association. For example, a study in over 1600 children of 4–18 years from UK showed no link between eating dinner after 20:00 and excess body weight [18]. Differences among studies could be related to the definition of late eating that in our work was based on the median values and established one hour later (~21:00) in the current Spanish population, than in the aforementioned study (in UK). Cultural differences among countries could also explain differences [9,10]. Spain is located relatively westward within its time zone, resulting in sun rise and sun set occurring at a later time as compared to many other countries within the same time zone, such as Germany. The late meal timing in Spain is thus less extreme as compared to solar time, and thus more comparable to other countries than what the clock time of the behavioural cycle, including eating, suggests. For this reason, the concept and the results of this study are likely generalizable to many other countries with different eating timings. Regardless, when translating to other countries, and to account for any systematic differences between countries, we suggest that it may be most informative to categorize meal timing according to the median value of each population (as we have considered here).

Several factors could also account for the association between late eating and obesity found in the current population (especially among girls), such as sleep duration [32]. In the current school children population, non-significant differences were found between LDE and EDE in sleep duration as objectively assessed. However, the time in bed (subjectively assessed) was shorter in LDE than EDE. Nevertheless, when the association between late eating and obesity was adjusted for the objective sleep duration, significance was maintained, suggesting that late eating relates to obesity, independently of sleep characteristics.

The detrimental impact of excess body weight in children is largely related to chronic low-grade inflammation [33]. However, there are no studies about the potential impact of dinner timing on inflammatory markers in children. Our results show that those children who had a late dinner had 1.4-fold CRP and 1.6-fold IL-6 values than early eaters. These two pro-inflammatory biomarkers have been related to BMI and have been involved in the pathogenesis of obesity [34,35], which suggests a deleterious association of late eating with inflammatory markers at these ages. Interleukine-6 (IL-6) is the main regulator of the acute inflammatory response. It is a pleiotropic cytokine that plays a critical role in chronic inflammation by stimulating the synthesis of CRP [36]. CRP is currently an extensively utilized biomarker for monitoring inflammation in the paediatric and neonatal populations [37], and children with metabolic syndrome have shown to be approximately three times more likely to have an elevated CRP level than those without metabolic syndrome [38]. Nevertheless, it is unknown whether children with an elevated CRP level are more likely to experience cardiovascular complications in the future.

Cortisol is known as the stress hormone. However, it has many more functions, including important regulatory effects throughout the body and brain, impacting energy and metabolic processes and immune and inflammatory system functioning, among others [39]. Furthermore, cortisol is one of the more relevant circadian hormones, and it is considered to be a marker of the internal clock, although behavioural and environmental factors also exert an influence [40].

Our data show that late dinner eaters had significantly lower values in the amplitude of the daily patterns of cortisol and of peripheral body temperature. The lower amplitude values in LDE may be an indicator of circadian system alterations in these children who dine late [19,41]. Nevertheless, the lower values in cortisol at breakfast time may be also related to the later circadian phase that characterizes late eaters. At those hours, cortisol levels may have not reached the morning peak yet. For these children, the intake of breakfast around 09:00 probably occurs at an earlier circadian phase, i.e., toward the biological night, which may also affect metabolism [42], although future studies with higher temporal resolution of cortisol sampling across 24 h under circadian protocols are required to distinguish changes in timing vs. amplitude of rhythmicity and to distinguish circadian vs. behavioral contributors to these changes.

In any case, the lower levels of morning cortisol in LDE than in EDE could be related to a lower appetite during breakfast [43], and may partly explain the lower breakfast energy intake that characterizes late eaters. This observation, in part, could explain the higher BMI of the LDE group, since poor breakfast or skipping breakfast has been associated with obesity in both adults and children [44,45]. This lower hunger and appetite in the early biological morning and the later dinner, might also relate to a delay in the circadian rhythm in hunger and appetite relative to clock time [46]. Food intake is considered to be one of the main synchronizers of peripheral oscillators [47] and late eating may lead to an uncoupling of these biological clocks [48] inducing metabolic alterations in children [5,48]. Indeed, in the current population, LDE had a delay in the temperature increase after dinner, which may be capturing a delay in the peripheral clocks [49] and/or an acute effect of the delayed dinner intake. In addition, the significantly lower temperature values in the postprandial hours in dinner of LDE as compared to EDE, may be related to a reduced diet induced thermogenesis (DIT), and be involved in the differences found in body weight between late eaters and early eaters [50,51,52].

In the discussion of our results, we need to consider some limitations. Although most of the prior published studies of timing in food intake have been carried out through questionnaires, and we need to use the same methodology to compare the findings, seven-day dietary records are subjective methods. Furthermore, they were not entirely completed by some children (*n* = 35 out of 432). This is particularly problematic in children and adolescents, since children may be more likely to forget to register food consumption. We suggest replacing paper recordings by a simpler and more attractive tool such as a smartphone application to capture the moment of food intake (e.g., photos of food intake) [10,53]. In this observational study, causality cannot be established. Furthermore, considering that many exposures and outcomes were correlated with each other, we did not correct for multiple comparisons, therefore more future studies are needed to confirm our findings.

## 5. Conclusions

This study represents a significant step towards the understanding of novel aspects in timing of food intake and its relationship with chronodisruption and metabolic risk in children. Future studies in school-age children are needed to test if advancing the dinner timing can improve metabolic, inflammatory and circadian-related alterations and prevent obesity.

## Figures and Tables

**Figure 1 nutrients-12-02881-f001:**
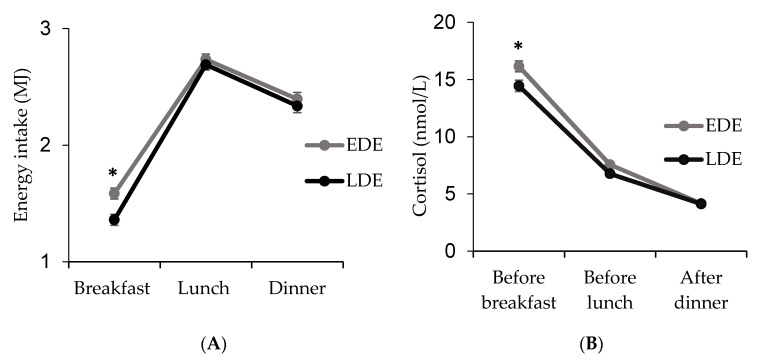
Distribution of energy intake across meals (**A**) and daily pattern of cortisol (before breakfast (09:00), before lunch (14:00), after dinner (23:00)) (**B**) in Late Dinner Eaters (LDE) and Early Dinner Eaters (EDE). (*): Differences statistically significant (**A**) *p* = 0.002; (**B**) *p* = 0.044.

**Figure 2 nutrients-12-02881-f002:**
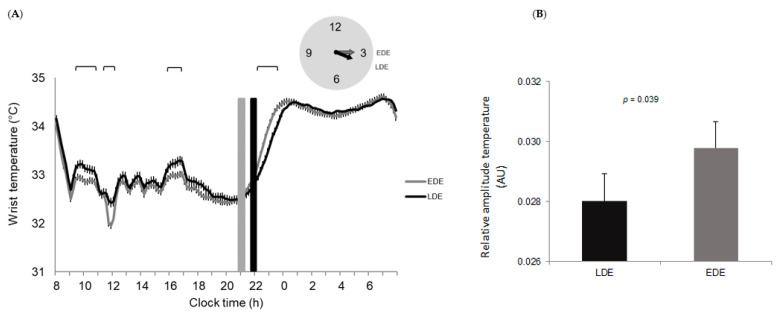
Daily patterns of wrist temperature in Late Dinner Eaters (LDE) and Early Dinner Eaters (EDE) are represented in (**A**). The upper brackets represent the hours at which the pattern differs significantly (*p* < 0.05). Vertical bars represent dinner timing and duration (width of the bar) in EDE (in gray) and LDE (in black). The clock represents the different temperature acrophases of EDE (in gray) and LDE (in black). Differences in relative amplitudes of temperature between LDE and EDE are represented in (**B**).

**Table 1 nutrients-12-02881-t001:** General characteristics of total, Late Dinner Eaters (LDE) and Early Dinner Eaters (EDE) children.

	Total	SD	LDE	SD	EDE	SD	*p*
General characteristics							
N	397		197		200		
Female (%)	50.7		52.5		47.5		0.272
Overweight or Obesity (%)	30.5		37.1		24		**0.003**
Age (y)	10	1.2	10	1.2	10	1.2	0.159
Weight (kg)	41.4	12.2	42.9	10.9	38.6	11.6	**<0.001**
Height (m)	1.45	10.4	1.47	9.9	1.43	10.4	**<0.001**
BMI (kg/m^2^)	19.4	3.9	19.6	3.5	18.7	3.8	**0.004**
BMI z-score	1.1	2.1	1.4	2.1	0.8	2.1	**0.003**
Body fat (%)	21.2	7.5	22.0	7.3	19.5	7.2	**0.002**
Waist circumference (cm)	65.4	9.9	66.6	9.4	63.5	9.6	**0.004**
Timing of food intake							
Breakfast (h)	08:33	0:27	08:35	00:29	08:31	00:24	0.109
Lunch (h)	14:24	0:19	14:24	00:19	14:21	00:17	0.084
Dinner (h)	21:07	0:31	21:31	00:19	20:43	00:18	**<0.001**
Midpoint of food intake (h)	14:49	0:21	15:03	00:17	14:37	00:16	**<0.001**
Bedtime (h)	22:49	0:39	23:04	00:34	22:36	00:35	**<0.001**
Get up time (h)	08:13	0:29	08:17	00:29	08:08	00:27	**0.001**
Food intake							
Breakfast (% of daily energy)	17.2	6.4	16.3	4.8	18.4	7.5	**0.002**
Second breakfast (% of daily energy)	10.6	4.2	10.7	4.3	10.5	4.2	0.672
Lunch (% of daily energy)	32.6	5.9	32.8	5.6	32.2	6.2	0.338
After lunch (% of daily energy)	12.1	5.5	12.5	5.5	11.8	5.4	0.267
Dinner (% of daily energy)	27.7	6.4	28.0	6.0	27.4	6.7	0.384

BMI: Body Mass Index; SD: Standard deviation. Significant differences are represented in boldfaces.

**Table 2 nutrients-12-02881-t002:** Differences in metabolic and inflammatory markers in saliva between Late Dinner Eaters (LDE) and Early Dinner Eaters (EDE).

Inflammatory Markers	TOTAL			LDE			EDE			
	Median	5%	95%	Median	5%	95%	Median	5%	95%	*p*
Glucose mg/dL	3.6	0.1	58.7	2.9	0.1	70.3	4.9	0.1	58.5	0.352
Insulin pg/mL	12.1	3.8	107.1	9.8	3.8	169.7	12.1	3.8	103.3	0.413
Triglycerides mg/dL	1.3	0.7	24.0	1.2	0.7	43.1	1.7	0.7	24.7	0.691
Leptin pg/mL	20.1	19.0	22.0	20.1	19.0	22.9	19.0	19.0	22.5	0.861
CRP ng/mL	3.2	0.2	42.7	4.4	0.2	42.7	1.8	0.2	44.3	**0.009**
IL1b pg/mL	10.4	1.2	125.3	14.2	1.7	135.3	9.8	1.2	123.1	0.173
IL6 pg/mL	0.9	0.4	8.4	1.1	0.4	14.1	0.9	0.4	6.8	**0.036**
IL8 pg/mL	40.1	4.6	299.4	40.7	5.1	309.3	40.1	6.1	322.0	0.708
TNFα pg/mL	0.7	0.3	4.8	0.7	0.3	5.1	0.7	0.3	3.9	0.876
MCP1 pg/mL	44.6	12.1	222.1	44.6	15.4	331.1	41.3	8.4	216.5	0.802
NGF pg/mL	0.5	0.3	0.7	0.5	0.3	0.8	0.5	0.3	0.6	0.877
HOMA-IR	0.4	0.1	9.9	0.4	0.1	17.7	0.2	0.1	9.2	0.448

Data are represented with the median and confidence interval. Mann–Whitney *U* test, between LDE and EDE was used. CRP: C-Reactive Protein; IL: Interleukin; TNFα: Tumor Necrosis Factor; MCP1: Monocyte Chemoattractant Protein 1; NGF: Nerve Growth Factor; HOMA-IR: Homeostasis Model Assessment of Insulin Resistance. Significant differences are represented in bold (*n* = 129).

**Table 3 nutrients-12-02881-t003:** Differences between Late Dinner Eaters (LDE) and Early Dinner Eaters (EDE) in the individual chronotype, sleep duration and dinner characteristics.

	LDE		EDE		
	*n* = 197		*n* = 200		
	Mean	SEM	Mean	SEM	*p*
Total Energy Intake (MJ/day)	8.3	0.1	8.6	0.1	0.141
Midpoint of food intake	15:03	00:01	14:37	00:01	**<0.001**
Dinner duration (min)	0.28	0.01	0.30	0.01	**0.043**
Time in bed (h)	9.1	0.04	9.5	0.04	**<0.001**
Objective sleep duration (h)	7.8	0.07	7.7	0.07	0.244
Chronotype (sleep centre; MCTQ) (hh:mm)	4:09	0:02	3:54	0:02	**0.001**
Evening-types (%)	4.2		3.0		**0.026**
Average activity (°/min)	46.3	0.4	46.7	0.4	0.503

Data were analysed by ANCOVA, adjusted by gender, schools, age and academic year. MCTQ: Munich Chronotype Questionnaire; SEM: Standard Deviation of the Mean. Significant differences are represented in bold.

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
