# Peer review of "Late Eating Is Associated with Obesity, Inflammatory Markers and Circadian-Related Disturbances in School-Aged Children"

_nutrients, 2020, doi:10.3390/nu12092881_

Round 1

Reviewer 1 Report

  1. BMI is not the best measure of adiposity in children as it can not be compared across gender and age. As a result standardized BMI should be used of percentage of 95th percentile.
  2. Why were parent diaries used to measure sleep when children wore a tri-axial acclerometer? I would have been a stronger design to use the sleep diaries to inform the identification of sleep episodes in the accelerometer data.
  3. What time of day were the saliva samples collected to measure metabolic and inflammatory markers? Was this standardized across participants?
  4. What diet coded by a trained dietitian, did anyone review the coding of the diet data? Why were breakfast and dinner selected as the endpoints of the day rather than the first and last eating episode of the day? Did children snack after dinner? How were bedtime snacks handled? How did you ensure that diet records were completed by children on a daily basis and not right before turning them in? Did parents review the records? Were children trained to identify serving sizes?
  5. What criteria were used to determine the accelerometer and temperature sensor were worn and that accurate data were obtained?
  6. Authors should control for multiple testing.
  7. How many children had complete data or missing data. How were missing data handled? 
  8. Table 1. It would be helpful to add these descriptives for the LDE and EDE groups (in addition to the overall sample data)
  9. Include the sample size of participants included in Table 3. Did you have complete data on all of these kids?
  10. Context should be added regarding local time in Spain not well aligned with sun time resulting in later clock times of eating, relative to many other countries.
  11. Limitations, the authors acknowledge limitations associated with self report dietary intake, but should also mention that this is particularly problematic in children and adolescents.
  12. Line 314 If children always eat late, then can they really be disrupted? If they are consistent in their behavior, their phase is later, but not disrupted.
  13. Can you add statistics on the intraindivdual variation of meal timing and the midpoint of food intake?
  14. Can normative data on children's week day and weekend meal timing be added in a supplemental table by age and maybe gender? This might help researchers planning future studies in this area. Do you observe social jet lag in the midpoint of food intake and does this increase with age?
  15. Use of self reported sleep should also be noted as a limitation.
  16. Waist is unclear. Waist circumference would be more better. Also provide additional information on how this was measured.

Author Response

Dear Editor,

We would like to give thanks to the reviewers for the constructive feedback and for the opportunity to further improve and resubmit our work entitled: “Late eating is associated with obesity, inflammatory markers and circadian-related disturbances in school-aged children” with manuscript number Submission 917823.

We have addressed each of the reviewer comments in a point-by-point fashion. The changes made in the revised manuscript have been highlighted in yellow colour for clarity.

Thank you again and we are looking forward to hearing from you.

Yours sincerely,

Dr. Marta Garaulet

REFEREE 1

 1. BMI is not the best measure of adiposity in children as it can not be compared across gender and age. As a result standardized BMI should be used of percentage of 95th percentile.

-Answer:We thank the reviewer for this very important insight. Now we have transformed the BMI to BMI z-score by WHO Growth Reference [World Health Organization. WHO Child Growth Standards: Length/Height-for-age, Weight-forage, Weight-for-length, Weight-for-height, and Body Mass Index-for age: Methods and Development. 1st ed. World Health Organization; 2006] and performed the analyses again. The results remained the same for Z-BMI between Late Dinner Eaters (LDE) and Early Dinner Eaters (EDE) P=0.003. We have added this new information in Table 1 and materials and methods section (page 3, line 122).

2. Why were parent diaries used to measure sleep when children wore a tri-axial acclerometer? I would have been a stronger design to use the sleep diaries to inform the identification of sleep episodes in the accelerometer data.

-Answer: we agree that objectively estimated sleep duration from the accelerometer can be a more accurate measure for sleep duration than self-reported. Thus, in the revision, we have re-named previous “sleep duration variable” as “time in bed”, and we have included a new variable of sleep duration as objectively measured and results have been incorporated into Table 3.

We have performed further adjustments with both variables and we obtained that the association with obesity was maintained when we adjusted for objective sleep duration.

We have also introduced the same modification in the names of the variables in the Methods section accordingly: “Children (together with their parents) completed 7-day sleep diaries adapted for the appropriate age group (25) which included: a) time in bed (bedtime, number of awakenings during sleep and get up time); b) time and duration of naps (20). Furthermore, sleep duration was objectively determined by a formula integrating the objectively measured motor activity, body position and skin temperature as previously described (26). (Page 3, lines 131-135).

Moreover, in Discussion we have modified: "Several factors could be also accounting for the association between late eating and obesity found in the current population (especially among girls) such as sleep duration (32). In the current school children population non-significant differences were found between LDE and EDE in sleep duration as objectively assessed. However, the timing in bed (subjectively assessed) was shorter in LDE than EDE. Nevertheless, when the association between late eating and obesity was adjusted for the timing in bed or the objective sleep duration, significance was maintained suggesting that late eating relates to obesity independently of sleep characteristics.” (Page 10, line 326-332).

  1. Azevedo C, Sousa I, Paul K, MacLeish M, Mondejar M, Sarabia J, et al. Teaching chronobiology and sleep habits in school and university. Mind, Brain, and education. 2008;2(1):34-47.
  2. Garaulet M, Madrid JA. Methods for monitoring the functional status of the circadian system in dietary surveys studies: application criteria and interpretation of results. Nutr Hosp. 2015;31 Suppl 3:279-89.
  3. Ortiz-Tudela E, Martinez-Nicolas A, Campos M, Rol MA, Madrid JA. A new integrated variable based on thermometry, actimetry and body position (TAP) to evaluate circadian system status in humans. PLoSComput Biol. 2010;6(11):e1000996.
  4. Li L, Zhang S, Huang Y, Chen K. Sleep duration and obesity in children: A systematic review and meta-analysis of prospective cohort studies. J Paediatr Child Health. 2017;53(4):378-85.
  1. What time of day were the saliva samples collected to measure metabolic and inflammatory markers? Was this standardized across participants?

-Answer: In order to measure metabolic and inflammatory markers, saliva samples were collected in a standard centrifugation tube with a small cotton swab inside (Salivette; Sarstedt, Barcelona, Spain). Samples were collected at home with their parents help, in the morning before breakfast at 09:00h (n=129) in fasting conditions, the same day of the week (Sunday) for all the children as previously done in adults. This information has been added in the revised manuscript in the Methods section (page 4, line 140).    

  1. What diet coded by a trained dietitian, did anyone review the coding of the diet data? Why were breakfast and dinner selected as the endpoints of the day rather than the first and last eating episode of the day? Did children snack after dinner? How were bedtime snacks handled? How did you ensure that diet records were completed by children on a daily basis and not right before turning them in? Did parents review the records? Were children trained to identify serving sizes?

-Answer: We thank the reviewer for pointing this out. The 7-day dietary records, including food quantities and timing, were completed daily by the children with their parent´s help. They were also informed that they had to weigh the food and write it down every day. Furthermore, the staff was in constant communication with the parents through a cell phone app for any enquiries or concerns that could arise during the intervention week. The diets were coded by a trained dietitian and reviewed by a senior nutritionist and total energy intake and macronutrients composition was analysed with a nutritional evaluation software program (Grunumur 2.0 8) (21) based on Spanish food composition tables (22). (Page 3, line 106-113).

  1. Perez-Llamas F, Garaulet M, Torralba C, Zamora S. [Development of a current version of a software application for research and practice in human nutrition (GRUNUMUR 2.0)]. Nutr Hosp. 2012;27(5):1576-82.
  2. Moreiras O, Carvajal A, Cabrera L. (Table of Composition of Spanish Foods) Tablas De Composición De Alimentos (In Spanish). 1995.
  1. What criteria were used to determine the accelerometer and temperature sensor were worn and that accurate data were obtained?

-Answer: We thank the reviewer for the comment. We have added explanations addressing accelerometry data reduction, validation, and the criteria to determine wear vs. non-wear time in the Methods section. To facilitate the review, we have copied the text here: “In brief, motor activity was expressed as the accumulative changes in three-axis tilt with respect to the previous point and expressed as degrees per minute. Body position was calculated as the angle between X-axis of the accelerometer and the horizontal plane, where 0° represents the arm in a horizontal position and 90° when vertical.  This method has been previously validated with the commonly used wrist-worn Actiwatch accelerometer, and almost all the parameters derived from the rhythm showed high correlations between both devices (31). Because the device used in the current study integrates the activity sensor together with a temperature sensor, we were able to use both the motor activity and temperature readings to estimate wear time. Non-wear time was defined as an interval in which motor activity readings were lower than 4 º/min and the skin temperature readings were out of the physiological range (i.e., <28ºC). A valid day was defined as a day with at least 16 hours of wear time.” (Page 4, lines 164-175).

  1. Bonmati-Carrion MA, Middleton B, Revell VL, Skene DJ, Rol MA, Madrid JA. Validation of an innovative method, based on tilt sensing, for the assessment of activity and body position. Chronobiol Int. 2015;32(5):701-710.
  1. Authors should control for multiple testing.

-Answer: We understand the reviewer’s concern. In order to decrease type 1 error, in the original design of the study, BMI was selected as the primary outcome, while the pro-inflammatory markers were considered as secondary outcomes. We have indicated this now clearly in the revised manuscript (page 11, lines 381-383). Furthermore, corrections may not be urgent because the traits included in the analyses were highly correlated. In addition, we are a bit hesitant to correct for multiple comparisons, since it is at the cost of an increasing false negative (type 2 error). This may let us miss out on important discoveries, especially considering that such studies on food timing and health outcomes in pediatric populations are rare and require facing many obstacles to be performed. Nevertheless, to address the reviewer’s concern, we have mentioned this as a limitation in the Discussion and stated that “more future studies are needed to confirm our findings”.

  1. How many children had complete data or missing data. How were missing data handled?

-Answer: we thank the reviewer for this comment. In the original sample, out of a total of 432 children, a total of 397 children had fully completed data and only 35 children did not complete the 7-day dietary records. Therefore, these 35 children were initially excluded from the study (page 11, line 376). This is the reason why the population sample consists of 397 children. As for the inflammatory markers, determinations were performed only in 129 children, due to budget limitations (page 4, line 144).

  1. Table 1. It would be helpful to add these descriptives for the LDE and EDE groups (in addition to the overall sample data)

-Answer: We thank you for your insight on this table. We have incorporated the descriptive data as suggested in new Table 1. We have fully removed the old Table 2.

  1. Include the sample size of participants included in Table 3. Did you have complete data on all of these kids?

-Answer: Thank you so much for your comments, we have added “n = 129” in Table 2 legend.

  1. Context should be added regarding local time in Spain not well aligned with sun time resulting in later clock times of eating, relative to many other countries.

-Answer: we agree with the reviewer´s suggestion. It is true that Spain is located relatively westward within its time zone, resulting in sun rise and sun set occurring at a later time as compared to many other countries within the same time zone, such as Germany. The late meal timing in Spain is thus less extreme as compared to solar time, and thus more comparable to other countries as compared to what the clock time of the behavioral cycle, including eating, suggests. For this reason, the concept and the results of this study are likely generalizable to many other countries with different eating timings. Regardless, when translating to other countries, and to account for any systematic differences between countries, we suggest it may be most informative to categorize meal timing according to the median value of each population (as we have considered here).We have provided this extra context regarding local time in Spain in the Discussion (page 10, lines 317-325).

  1. Limitations, the authors acknowledge limitations associated with self report dietary intake, but should also mention that this is particularly problematic in children and adolescents.

-Answer: Thank you so much for your comments, we have added “this is particularly problematic in children and adolescents since children may be more likely to forget to register food consumption” in the Discussion section (page 11 and line 376-378).

  1. Line 314 If children always eat late, then can they really be disrupted? If they are consistent in their behavior, their phase is later, but not disrupted.

-Answer: This is an interesting point. We agree ‘disrupted’ could be interpreted as meaning ‘irregular’, which is not what we intended. We used it to as a broad term encompassing late timing. Indeed, while the timing of eating behaviour may be very consistent and regular from day to day, it may still be timed at an adverse central circadian phase, i.e., compared to markers of the central circadian pacemaker, such as the profile of melatonin under dim light conditions, or—to some extent—to that of cortisol. Indeed, we have shown that the timing of eating shows a better correlation with BMI and adiposity in college students when expressed as the circadian timing of eating (i.e., timing relative to the dim light melatonin onset [DLMO]) as compared to the clock timing of eating (1). In summary, we agree with the reviewer that the word ‘disrupted’ may be too vague and not accurately describe the late timing of eating that is the topic of the current work. For this reason, we have removed the word ‘disrupted’, and instead refer to the timing of eating per se.  

  1. McHill AW, Phillips AJ, Czeisler CA, et al. Later circadian timing of food intake is associated with increased body fat. Am J Clin Nutr. 2017;106(5):1213-1219. doi:10.3945/ajcn.117.161588
  1. Can you add statistics on the intraindividual variation of meal timing and the midpoint of food intake?

-Answer: Thank you we have added this information in the revised manuscript, in Supplementary table 2, in Methods (page 3, line 98-105) and Results (page 6, line 222). However, no differences were found in the intraindividual variation of dinner timing and the midpoint of food intake.

  1. Can normative data on children's week day and weekend meal timing be added in a supplemental table by age and maybe gender? This might help researchers planning future studies in this area. Do you observe social jet lag in the midpoint of food intake and does this increase with age?

-Answer: Thank you,we have added this information to the revised manuscript (Supplementary table 3) and  to the Results section: “Social jet lag of dinner timing and midpoint of food intake differed across school grades with grade 2 (10-11 ages) having less social jet lag than grades 1 (8-9 ages) and 3 (11-12 ages). No significant differences were found in social jet lag values between sexes (Supplementary table 3).” (Page 6, line 225-228). We also have included the definition of social jet lag of dinner timing and midpoint of food intake in Methods (page 3, lines 98-105).

  1. Use of self reported sleep should also be noted as a limitation.

-Answer: we have included objective measurements in the Methods section as well as in the Results. Page 3, line 131-135. 

  1. Waist is unclear. Waist circumference would be more better. Also provide additional information on how this was measured.

-Answer: We thank the reviewer for the suggestion and we have changed “waist” by “waist circumference” in the revised manuscript (table 1). Waist circumference was measured in standing position with their feet together at the midpoint between the last rib and the crest of the ilium (top of the hip bone). Measurements were done after participants placed their arms at their side with the palms of their hands facing inwards and breathing out gently. Data were recorded to the nearest 0.1 cm using a metallic tape. And this information has been included in the revised manuscript (page 3, lines 123-127).

Thank you very much for the suggestion. We have verified that the parts with a high similarity degree correspond to the Methods section of other studies that we had previously performed over the same population (ONTIME-Jr). This population has been extracted from our database ONTIME and previous analyses have been done and used for other papers. This is why the methodology we describe is very similar between papers.

Reviewer 2 Report

Thank you very much for this well written paper. It is an interesting topic in times of discussions about meal skipping or intermitted fasting.

I have just some few comments:

Line 105 did the authors also used waist circumference percentiles to categorize the obesity status of children?

Did the authors overserve any sex-specific differences for all data? If no please present this information also in the results section.

Did the authors have any information about the LDE and EDE behavior in other countries for children? The dinner time in Spain is different from northern countries (earlier dinner time), thus in consequence could the results from the present study be comparable to other countries?

Author Response

Dear Editor,

We would like to give thanks to the reviewers for the constructive feedback and for the opportunity to further improve and resubmit our work entitled: “Late eating is associated with obesity, inflammatory markers and circadian-related disturbances in school-aged children” with manuscript number Submission 917823.

We have addressed each of the reviewer comments in a point-by-point fashion. The changes made in the revised manuscript have been highlighted in yellow colour for clarity. 

Thank you again and we are looking forward to hearing from you.

Yours sincerely,

Dr. Marta Garaulet

REFEREE 2
Thank you very much for this well written paper. It is an interesting topic in times of discussions about meal skipping or intermitted fasting.
I have just some few comments:
1. Line 105 did the authors also used waist circumference percentiles to categorize the obesity status of children?

-Answer: We thank the reviewer for the comment. Children were categorised into normal weight and overweight/obese according to the sex-and-age specific BMI cut-points proposed by the International Obesity Task in the revised version of the manuscript (23). This has been incorporated into page 3, line 122.
23. Cole TJ, Flegal KM, Nicholls D, Jackson AA. Body mass index cut offs to define thinness in children and adolescents: international survey. BMJ. 2007;335(7612):194.

2. Did the authors overserve any sex-specific differences for all data? If no please present this information also in the results section.
-Answer: Thank you so much for your comments. We have included a new table showing the differences in BMI and waist circumference between Late Dinner Eaters (LDE) and Early Dinner Eaters (EDE) in both genders as “Supplementary table 1”and we have added in Results: “When standardizing by sex, these associations remained unchanged only in girls (Supplementary table 1).” (Page 6, line 221). 

3. Did the authors have any information about the LDE and EDE behavior in other countries for children? The dinner time in Spain is different from northern countries (earlier dinner time), thus in consequence could the results from the present study be comparable to other countries?

-Answer: we agree with the reviewer’s suggestion. It is true that Spain is located relatively westward within its time zone, resulting in sun rise and sun set occurring at a later time as compared to many other countries within the same time zone, such as Germany. The late meal timing in Spain is thus less extreme as compared to solar time, and thus more comparable to other countries as compared to what the clock time of the behavioral cycle, including eating, suggests. For this reason, the concept and the results of this study are likely generalizable to many other countries with different eating timings. Regardless, when translating to other countries, and to account for any systematic differences between countries, we suggest it may be most informative to categorize meal timing according to the median value of each population (as we have considered here). We have provided this extra context regarding local time in Spain in the Discussion (page 10, lines 317-325).

Thank you very much for the suggestion. We have verified that the parts with a high similarity degree correspond to the Methods section of other studies that we had previously performed over the same population (ONTIME-Jr). This population has been extracted from our database ONTIME and previous analyses have been done and used for other papers. This is why the methodology we describe is very similar between papers.
Nevertheless, we have modified the Methods sections to decrease similarity.

Please write down
